# MPLNet: Mamba prompt learning network for semantic segmentation of remote sensing images of traditional villages

Cheng Zhang[1,2], PeiLin Liu[1,2], JinLin Teng[1,2], Chunqing Liu[1,2]*

**1** College of Landscape Architecture and Art, Jiangxi Agricultural University, Nanchang, China, **2** Jiangxi Rural Culture Development Research Center, Nanchang, China

* liuchunqing@jxau.edu.cn

## Abstract

In recent years, the study of semantic segmentation of remote sensing images (RSI) has gained significant attention due to its critical role in geospatial analysis, agriculture, and forestry. However, existing remote sensing segmentation methods face several challenges: (1) limited dataset diversity and inadequate exploration of traditional village landscapes, resulting in a lack of geospatial representation for these unique environments; (2) inefficiencies in same-layer or cross-layer feature fusion when using convolutional neural networks (CNNs) or transformers, leading to either insufficient spatial modeling or excessive computational demands; and (3) multimodal approaches that improve modeling accuracy but introduce high parameter complexity and computational overhead. To address these issues, we propose the Mamba Prompt Learning Network (MPLNet) for efficient and accurate RSI segmentation, with a strong emphasis on spatial information extraction and GIS-based applications. First, we construct TV-RSI, a highly diverse large-scale data set specifically designed to capture the spatial structures, topographic variations, and land use patterns of traditional villages. Second, we develop the Mamba Fusion Module, which improves geospatial feature utilization by efficiently modeling both intralayer and interlayer spatial relationships, ensuring comprehensive feature extraction. Finally, we introduce prompt learning, which transfers bimodal geospatial knowledge from heavy-weight networks into a lightweight unimodal model, improving segmentation accuracy while maintaining computational efficiency. Extensive experiments on TV-RSI and two publicly available RSI datasets demonstrate that MPLNet achieves state-of-the-art performance with significantly reduced computational costs, making it an ideal solution for geospatial segmentation tasks in GIS-driven remote sensing applications.

**Data availability statement:** The datasets and code used in this study are publicly available for access and use by other researchers. The relevant data, including the Traditional Villages Remote Sensing Image (TV-RSI) dataset, which covers spatial information such as architectural distribution, topography, and land use, has been uploaded to the GitHub repository https://github.com/Jack13026212687/MPLNet). The repository also contains the implementation of the MPINet model, along with data preprocessing and training scripts, allowing researchers to reproduce the results presented in this study. The public availability of the data and code ensures long-term accessibility and facilitates further research in related fields. For any inquiries or additional requests, please contact the corresponding author.

**Funding:** We thank the National Natural Science Foundation of China for supporting this research through the projects "Gene Identification and Map Construction of Traditional Rural Settlement Landscapes in the Ganjiang River Basin" (Serial No. 51968026) and "Research on the Visual Perception, Quantitative Characterization, and Visual Evaluation of Traditional Village Landscape Resources in Ganjiang River Basin" (Serial No. 52268012). We also acknowledge the support of Jiangxi Rural Culture Development Research Center. We appreciate the technical assistance provided by the GIS and Remote Sensing Laboratory at Jiangxi Agricultural University. Special thanks to all members of the research team for their valuable discussions and contributions to the project.

**Competing interests:** No authors have competing interests.

## Introduction

Semantic segmentation is a computer vision and geospatial analysis task that aims to classify each pixel in an image by assigning a category label to accurately differentiate objects based on their shapes, spatial distributions, and locations. This process enables computers to not only identify objects within an image, but also understand their spatial relationships, making it a crucial tool for Geographic Information Systems (GIS) and remote sensing applications. In recent years, semantic segmentation has shown strong performance in traditional village analysis tasks, including land use classification, cultural landscape identification, preservation of rural heritage, ecological monitoring, and spatial planning [1–3]. However, in traditional village environments, the application of remote sensing semantic segmentation remains underexplored due to the lack of high diversity and spatially representative datasets. The TV-RSI data set addresses this gap by offering a comprehensive geospatial resource that captures the spatial structures, land use patterns, and ecological characteristics of traditional villages, allowing more precise spatial modeling, historical landscape analysis, and sustainable planning. With its detailed spatial annotations and large-scale coverage, TV-RSI provides a critical foundation for advancing GIS-driven deep learning models, facilitating fine-grained segmentation and geospatial intelligence in traditional village conservation and development.

With the acceleration of modernization, traditional villages are receiving more and more academic attention as cultural heritage and historical witnesses. Researchers have thoroughly explored the cultural, ecological and social values of traditional villages, emphasizing their importance in sustainable development. The location and landscape pattern of traditional villages show a deep understanding of the natural environment, such as topography, climate, and water resources, reflecting the harmonious coexistence of humans and nature [4], and provide a valuable reference for modern landscape design [5]. The introduction of artificial intelligence technology has significantly improved the protection and monitoring efficiency of the traditional landscape heritage of villages [6], while research based on culture-landscape genes provides a new theoretical framework for its protection and development [7]. In addition, low-altitude drone remote sensing brings new methods for geospatial data acquisition and landscape management [8], and multi-resolution feature fusion technology further improves the precision of landscape analysis [9]. These studies not only provide a solid theoretical foundation for the protection of traditional villages but also open up a practical path for their sustainable development in modern society.

Convolutional neural networks and the Transformer have been rapidly developed in recent years, and a large number of networks have been extracted. For example, the full convolutional architecture based on a convolutional network realizes end-to-end prediction at the pixel-level and improves the precision of semantic segmentation through the merging of features of multiple layers [10]. On this basis, the self-attentive three-stream TSNet network further combines RGB and depth features to achieve high-precision semantic segmentation results for indoor scenes [11]. In addition, ESANet demonstrated efficient RGB-D segmentation performance in

mobile robot scene analysis, combining the advantages of real-time and high precision [12]. In order to further improve segmentation accuracy, ACNet significantly improves RGB-D semantic segmentation by designing the complementary attention module to effectively fuse RGB and depth features [13]. With the introduction of the converter structure, the architecture based on the Swin converter and DCFAM decoder improves contextual information extraction in semantic segmentation tasks and improves the resolution recovery effect [14]. In the field of remote sensing image segmentation, progressive reconstruction networks have effectively improved segmentation accuracy through the synergistic action of multiscale features and depth separation modules, and have demonstrated strong fine segmentation capability [15]. However, these methods usually use convolution or a Transformer, leading to the problem of insufficient modeling or high computation.

Recently, prompt learning approaches have gained great utility in vision and language. For example, MPLe multimodal prompt learning significantly improves the model's generalization ability in downstream tasks by enhancing the consistency of the visual and linguistic representations of CLIP [16]. In this paper, we provide a systematic review of cue-based learning paradigms, focusing on the applications and methods of prompt learning in natural language processing [17]. In terms of improving the generalization performance of CLIP models, CoCoOp proposes a dynamic conditional cue optimization method, which further enhances the migration ability between categories of the model [18]. Meanwhile, CoOp optimizes the CLIP cue design using learnable vectors, improving the adaptability of the visual language model to image recognition tasks [19]. In addition, ProGrad optimizes the visual language model using cue-aligned gradient optimization, enabling the visual language model to show a stronger generalization ability in tasks of fewer samples and in the cross-domain [20]. Recent studies have further advanced deep learning architectures for remote sensing and multimodal perception tasks. **FDNet** proposed a dual-path cross-encoding framework that captures both spatial and temporal dependencies for precipitation nowcasting, achieving significant improvements in contextual feature learning and prediction accuracy through parallel encoding pathways and feature fusion mechanisms [21]. In addition, **DDFNet** introduced a dual-domain fusion network that integrates time and frequency-domain representations through a decoupled attention mechanism, demonstrating strong robustness and generalization across complex signal environments [22]. Building on this, recent RSI segmentation advances fall into three lines closely aligned with our setting: geometry-/boundary-aware decoding for contour fidelity and small-object recall, anisotropic/sparse attention for thin structures in high-resolution scenes, and state-space (Mamba) models that aggregate context in *linear time*. We adopt the third while complementing the first two via four-direction SS2D scans and a gated residual prior within the student stream (student-only inference), under a unified protocol—yielding consistent gains in Boundary-F1 and Connectivity [23–28]. However, there is still room for exploration in the purely visual domain as a way to further improve the performance of visual networks.

In order to solve the above problems, this paper proposes a Mamba prompt learning network for efficient and accurate remote sensing image segmentation. Specifically, we first construct a large-scale, diveRSI.y-rich traditional village remote sensing data set named TV-RSI. Then, we design the Mamba fusion module to fully exploit and utilize the complementary information by modeling the same-layer or cross-layer features. Finally, we introduce prompt learning to inject bimodal knowledge from heavy-weight networks into lighter unimodal networks in the form of cues, resulting in more streamlined and accurate models. Our contributions are mainly as follows:

(1) For the first time, we constructed a traditional village remote sensing dataset with large diversity and scale, called TV-RSI, and proposed a new Mamba prompt learning network, called MPLNet.

(2) A Mamba Fusion Module called MFM is designed, which deeply mines and efficiently integrates complementary information through modeling of same-layer or cross-layer features to maximize the use of information.

(3) Through prompt learning, the bimodal knowledge in the heavy-weight network is transformed into cue information and injected into the lightweight unimodal network, thus constructing a more streamlined and accurate network.

(4) It is shown on the proposed TV-RSI dataset and two publicly available datasets that our proposed MPLNet achieves state-of-the-art performance.

## Dataset

The Traditional Villages Remote Sensing Dataset (TV-RSI) is a spatially enriched collection of remote sensing images meticulously designed to capture the geospatial characteristics of traditional villages. Using Geographic Information System (GIS) technologies, TV-RSI systematically integrates multidimensional spatial data, including the distribution of village architecture, topographical variations, types of land cover, and surrounding ecological landscapes. This data set serves as a high-precision spatial repository, enabling comprehensive geospatial analysis, spatial pattern recognition, and terrain-based modeling of traditional village environments.One of the key spatial advantages of TV-RSI is its ability to analyze spatial relationships, detect Through high-resolution spatial analytics, researchers can quantify land cover transitions, assess urbanization impacts, and model the interdependencies between village structures and natural landscapes. The rich geospatial attributes of the data set enable users to perform fine-grained spatial assessments, such as village clustering, terrain-based segmentation, and proximity analysis of cultural heritage sites. In the context of traditional village conservation and sustainable development, the integration of remote sensing with GIS-based spatial modeling significantly improves spatio-temporal tracking, historical change analysis, and predictive mapping. TV-RSI provides a solid geospatial foundation for regional planning, heritage site management, and rural sustainability strategies, ensuring that conservation policies are data-driven and spatially optimized. Furthermore, its application in digital mapping, intelligent spatial analytics, and AI-enhanced geospatial modeling contributes to the advancement of digital heritage management and GIS-based decision support systems. By offering an unprecedented level of spatial granularity and thematic richness, TV-RSI is a valuable geospatial resource for researchers, planners, and policy makers. Bridges the gap between remote sensing, spatial intelligence, and the conservation of cultural heritage, fostering scientific advancements in geospatial analytics, spatial planning, and sustainable rural development.

The Traditional Villages Remote Sensing Dataset (TV-RSI) is a spatially comprehensive and high resolution dataset consisting of 77,850 images with a total volume of 6.6 GB , each at a resolution of $256 \times 256$ pixels, covering an extensive area of 166,900 square kilometers. Designed to support geospatial analysis and deep learning applications in GIS, TV-RSI systematically captures spatial structures, topographic features, and land use patterns unique to traditional villages. To ensure spatial diversity and analytical robustness, the data set is strategically partitioned into three subsets: the training set (68,984 images) for learning spatial relationships and geographic distributions, the validation set (7,627 images) for optimizing model generalization across diverse terrain and architectural configurations, and the test set (1,239 images) for evaluating model performance on complex spatial features and environmental variations. This spatially aware data set structure enhances the ability to perform precise land classification, spatial-temporal monitoring, and digital heritage mapping, providing a solid geospatial foundation for AI-driven GIS applications and sustainable development planning (Fig 1).

**Annotation protocol and QA.** High resolution ortho imagery was tiled into $256 \times 256$ patches (nonoverlapping), with rebalancing at sampling time to mitigate class skew. Six semantic classes (background, road, building, farmland, vegetation, drainage) were annotated at the pixel level using a standardized guideline. Each tile received double-pass verification by two annotators; conflicts were resolved by a senior reviewer. Spot audits in a random subset ensured consistency, particularly for ambiguous boundaries (e.g., etation-farmland edges). This protocol improves the quality of the labels without inflating the variance of the annotation.

Fig 2 illustrates the spatial distribution characteristics of different types of objects in the TV-RSI dataset, depicting the proportion of area (ranging from 0 to 1) occupied by categories such as background, road, building, farmland, vegetation, and drainage systems. The skewed distribution of background area proportions, predominantly concentrated in the 0.0-0.1 range, indicates that background elements occupy only a minor portion of most images, contributing to the spatial heterogeneity of the dataset. Similarly, roads and buildings exhibit a bias towards smaller spatial footprints, reflecting their localized and linear spatial characteristics in traditional village layouts. In contrast, farmland and vegetation show more uniform area distributions, with notable peaks in the 0.9-1.0 interval, suggesting that in certain images, these types of land

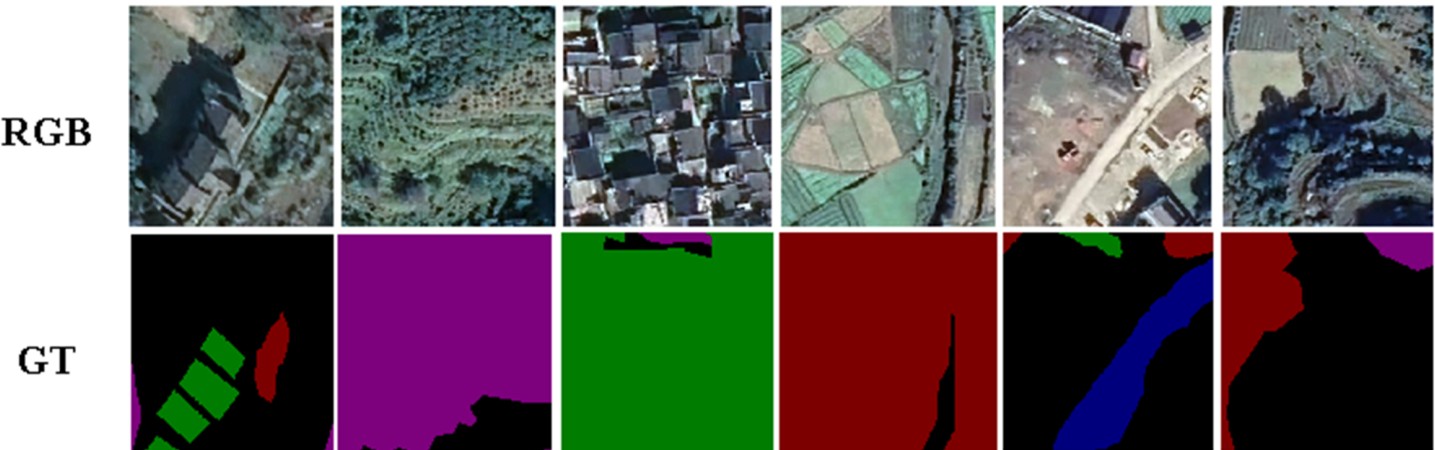

**Fig 1. Representative samples from the TV-RSI dataset.** All sampling, annotations, panel composition, and graphics are original works by the authors and are released under CC BY 4.0.

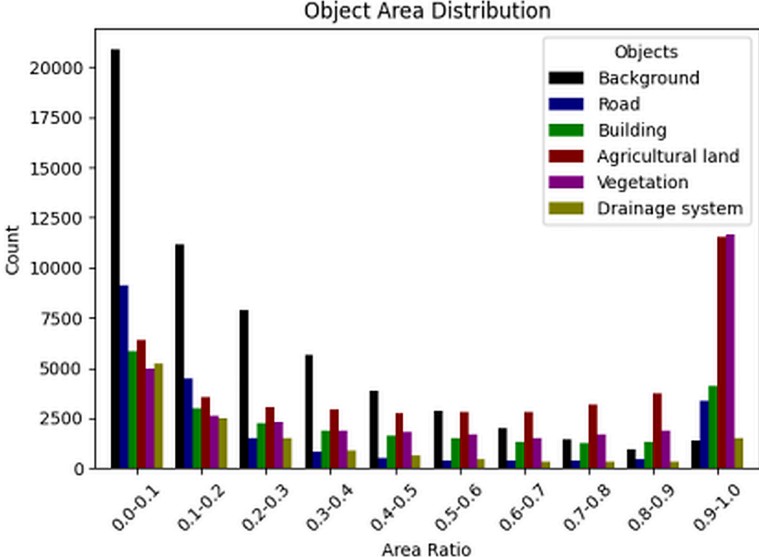

**Fig 2. Regional distribution of semantic objects.**

cover dominate the spatial composition. The dominance of small areas of roads, buildings, and background elements presents a unique spatial segmentation challenge, enriching the data set with diverse geospatial test cases for models targeting small-scale feature extraction. In contrast, extensive coverage of farmland and vegetation requires robust large-scale object recognition, making TV-RSI an ideal data set to evaluate multiscale spatial segmentation algorithms. By offering a rich spectrum of spatial AI-driven GIS applications in handling varied land cover distributions, geospatial complexities, and scale-aware segmentation tasks, the authors ultimately advance spatial intelligence in remote sensing analytics.

Fig 3 presents the spatial distribution heatmap of the location of objects in the TV-RSI dataset, which illustrates the variations in object density through a color gradient. The brighter yellow hues in the central region indicate a higher concentration of objects, while the color transitions to red and darker shades towards the edges, signifying a gradual decline in object density. This centrally aggregated spatial distribution highlights the tendency of key land cover elements - such as buildings, roads, and vegetation clusters - to be predominantly positioned in the core regions of images, while peripheral areas remain relatively sparse. This spatial characteristic provides crucial geospatial insights for GIS-driven segmentation models and spatial feature extraction, as it emphasizes key regions that warrant greater focus during model training. For location-sensitive remote sensing tasks, such as semantic segmentation and spatial object detection, the heat map offers a valuable spatial reference to enhance recognition accuracy and localization precision. Moreover, the structured spatial distribution of objects in TV-RSI supports targeted model optimization, facilitating the development of adaptive GIS-based deep learning algorithms that can efficiently process spatially clustered and dispersed land-cover patterns, ultimately advancing spatial intelligence in remote sensing analysis.

## Method

### Overall architecture

MPLNet consists of a **frozen bimodal teacher** and a **trainable unimodal student** (Fig 4). The teacher employs two ResNet-50 backbones for RGB ($R_1 - R_5$) and depth ($D_1 - D_5$), fused with Mamba Fusion Modules (MFM). Its parameters remain fixed during training to serve as a stable source of cross-modal context. The student is a single RGB backbone augmented with Mamba gates. During training, cross-stream knowledge is injected via *projection-aligned cues* (Eqs (1)–(5)). In inference, only the student stream is executed, preserving efficiency while retaining accuracy.

Fig 4 shows the overall structure of the proposed MPLNet. Specifically, the overall structure is divided into two parts, the upper and lower parts, labeled as the frozen part and the trainable learning part, respectively. In the freezing part, the backbone network adopts two symmetric ResNet-50s for the extraction of RGB and depth-modal information features,

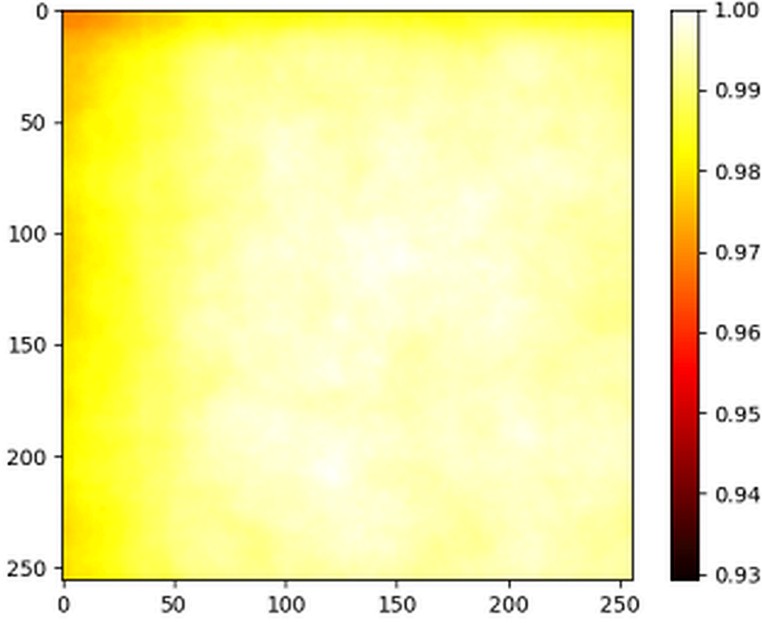

**Fig 3**. **Heat map of the object's position.**

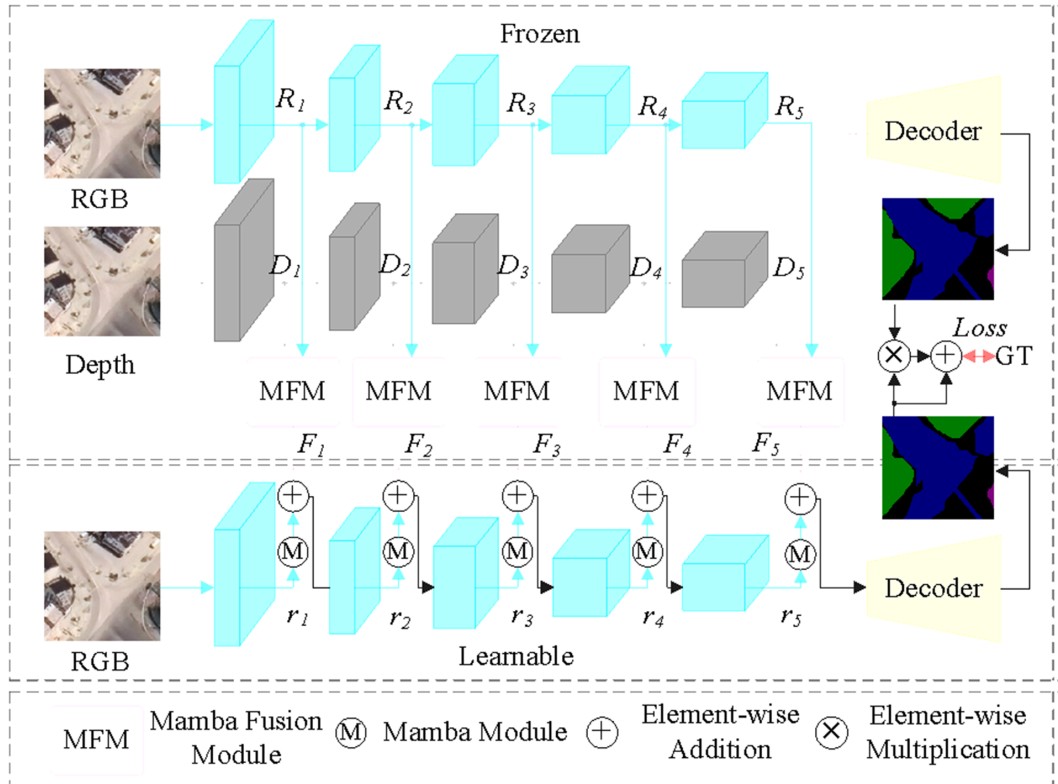

**Fig 4**. Overall architecture diagram of MPLNet.

respectively, followed by the efficient complementary utilization of multimodal information through the proposed MFM. The fused features are then the backbone network uses ResNet-50 for feature extraction of the RGB features, followed by efficient modeling of the features in context through Mamba blocks, and secondly, the enriched information from the frozen part is injected into the RGB features in the trainable learning part. Finally, the prediction map is obtained through the decoder and the output prediction map of the frozen part is injected as a cue to the output of the trainable learning part to obtain more accurate output results.

## MFM

In the feature fusion stage, a common approach is to perform modal feature fusion by employing either Convolutional Neural Networks or Transformers. However, Convolutional Neural Networks have insufficient global contextual modeling, and Transformers, although adequate in global contextual modeling, bring a large number of parameters and computational issues. To solve this problem, as shown in Fig 5, we introduce a Mamba mechanism for context-efficient modeling. Specifically, we first perform within-modal context modeling by Mamba for the two modalities, and then fuse the contextualized features through the ResNet-50 encoding layer to obtain more adequate complementary information. In addition, the Mamba mechanism obtains context in different directions through multiple nonlinear transforms and combines them by element-wise modulation to yield rich long-range cues. The Mamba mechanism performs sequential scans in four directions (left–right and top–bottom) on the 2D feature map, enabling the model to capture contextual dependencies from multiple orientations. It then fuses and smooths the results from all directions to integrate multi-directional information, enhancing the perception of boundaries and linear structures. The specific formulation is as follows.

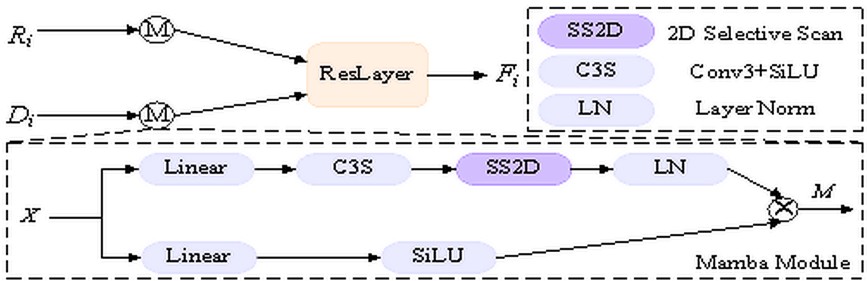

**Fig 5**. Structure of the Mamba Fusion Module (MFM).

**Design rationale and implementation details.** To make the operator pipeline explicit and reproducible, we formalize the intra- and intermodal processing used in MFM. Given a stage characteristic $X \in \mathbb{R}^{B \times C \times H \times W}$, we first apply a local mixing 3×3 $U = \text{Conv}_{3\times3}(X)$, then perform a 2D selective scanning (**SS2D**) along four directions $\{\rightarrow, \leftarrow, \downarrow, \uparrow\}$ to harvest long-range directional dependencies. The directional states are concatenated and projected back to the $C$ channels, followed by layer normalization.

$$S = \text{Proj}\left([S^\rightarrow \| S^\leftarrow \| S^\downarrow \| S^\uparrow]\right), \qquad Y = \text{LN}(S).$$

A lightweight 1×1 gate produces a content-adaptive mask $G = \text{SiLU}(W_m X)$ that modulates $Y$ through the element-wise product, producing $M(X) = G \odot Y$. Applying $M(\cdot)$ to the RGB and Depth streams separately gives $F_{r,i}$ and $F_{d,i}$, which are fused by a ResNet encoder block with 1×1 alignment $\phi(\cdot)$.

$$U = \text{Conv}_{3\times3}(X),$$
$$S^\rightarrow, S^\leftarrow, S^\downarrow, S^\uparrow = \text{SS2D}(U),$$
$$S = \text{Proj}\left([S^\rightarrow \| S^\leftarrow \| S^\downarrow \| S^\uparrow]\right), \tag{1}$$
$$Y = \text{LN}(S),$$
$$M(X) = \text{SiLU}(W_m X) \odot Y,$$
$$F_{r,i} = M(R_i), \qquad F_{d,i} = M(D_i), \tag{2}$$
$$F_i = \text{ResLayer}\left(\phi(F_{r,i}), \phi(F_{d,i})\right). \tag{3}$$

*Operator glossary and reproducibility notes.* $\text{Conv}_{3\times3}(\cdot)$ is a 3×3 convolution with same padding; $\text{SS2D}(\cdot)$ denotes 2D selective scan that aggregates directional state updates on the spatial grid; $\text{LN}(\cdot)$ is **Layer Normalization**; $\text{SiLU}(\cdot)$ is the SiLU activation; C3S is $\text{Conv}_{3\times3}(\cdot)$ and $\text{SiLU}(\cdot)$ operation; $\odot$ denotes element-wise multiplication; $\phi(\cdot)$ is a 1×1 projection for channel alignment; $\text{ResLayer}(\cdot)$ denotes a ResNet encoder block. In practice, SS2D adds linear time and memory in $HW$ and introduces negligible parameters compared to a ResNet-50 stage, which matches the efficiency requirements for remote sensing segmentation.

*Remark.* In our implementation, Conv3 refers to $\text{Conv}_{3\times3}$, SS2D is the 2D selective scanning operator, LN is layer normalization, SiLU is activation, and ResLayer corresponds to a ResNet-50 encoding block.

## Prompt learning strategies

The usual remote sensing semantic segmentation algorithms use multimodal encoding separately, followed by multimodal context fusion; however, this leads to a large number of parameters and computations in the inference phase, which is not

suitable in resource-constrained scenarios. To address this problem, we introduce a prompt learning mechanism. Specifically, in the training phase, first, we simultaneously train a multimodal bistream network and a single-stream unimodal network. Next, rich complementary information from the multimodal fusion is injected into the single-stream unimodal network by adding elements in the element way. Finally, the decoded output of the two-stream network is fused with the output of the single-stream network through a residual mechanism to obtain the final output. It is worth noting that in the inference stage, we only run the single-stream network as our final network, thus greatly reducing the number of parameters and computation, while obtaining better performance.

**Tensor shapes and reduction to the original form.** We use $R_i, D_i, T_i, r_i \in \mathbb{R}^{B \times C_i \times H_i \times W_i}$ and $P_i, Q_i \in \mathbb{R}^{B \times K \times H_i \times W_i}$, where $C_i$ is the stage-$i$ width and $K$ the number of classes. When $\sigma(\cdot) \equiv 1$ and layer normalization is omitted, Eq 4 reduces to the original update $f_i = T_i + M(r_i)$ (same semantics and implementation). For Eq 5, if $\sigma(x) = x$, $W_o = \mathbf{I}$, and $\tau_o = 1$, the gate becomes $1 + P_i$, hence $Q_i \leftarrow Q_i + P_i \odot Q_i = Q_i(1 + P_i)$. Introducing $\sigma$ and $\tau_o$ is only for numerical calibration and stability on class-imbalanced or high-confidence samples.

**Design rationale and implementation details.** Let $R_i, D_i$ denote the characteristics of the teacher (RGB/depth) in stage $i$, and let $T_i$ be the characteristic of the student in the same stage. We first build a *projection-aligned cue* by channel/spatial alignment $r_i = \phi([R_i \oplus D_i])$, where $\phi$ is a projection 1×1 (with up/down sampling if needed). To prevent gradient leakage from the student into the frozen teacher, we use a stop-gradient operator $\text{sg}(\cdot)$ on the cue during injection. A light gate, predicted from the student content, controls the injection strength. Formally, the student update (semantically equivalent to your $f_i = F_i + M(r_i)$) is written as

$$f_i = T_i + \sigma(W_g T_i) \odot \text{LN}\left(M(\text{sg}(r_i))\right) \tag{4}$$

where $M(\cdot)$ is the Mamba context operator from Eq 1, $\sigma$ is the sigmoid gate, $W_g$ is a projection 1×1, LN stabilizes the cue scale and $\odot$ is elemental modulation. When $\sigma(\cdot) \equiv 1$ and LN are omitted, Eq 4 reduces to the original $f_i = T_i + M(r_i)$; hence, the semantics and implementation remain unchanged while improving stability.

At the logit level, your multiplicative refinement $Q_i = P_i \times Q_i + Q_i$ is written in a numerically stable, confidence-gated form that preserves the same intent ("teacher prompts student") while avoiding uncontrolled logit growth:

$$Q_i \leftarrow (1 + \sigma(W_o P_i / \tau_o)) \odot Q_i \tag{5}$$

Here, $W_o$ is a projection 1×1 that aligns teacher records with student space, and $\tau_o > 0$ is a temperature controlling the sharpness of the gate. When choosing $\sigma(x) = x$, $W_o = \mathbf{I}$, and $\tau_o = 1$, Eq 5 becomes $Q_i + P_i \odot Q_i = Q_i(1 + P_i)$; introducing $\sigma$ and $\tau_o$ serves as numerical calibration and yields better stability under class imbalance and high-confidence samples. During inference, only the student branch (with gates) is executed, so the deployed model maintains unimodal complexity.

**Complexity.** All cue paths and the teacher decoder are training-only. The deployed model preserves the parameter count and FLOPs of the student, with negligible extra memory for the 1×1 gates.

### Loss function

We supervise both teacher and student with pixel-wise cross-entropy:

$$\mathcal{L}_{\text{teach}} = \sum_{i=1}^{5} \text{CE}(P_i, GT), \qquad \mathcal{L}_{\text{stud}} = \sum_{i=1}^{5} \text{CE}(Q_i, GT). \tag{6}$$

To encourage the student to absorb teacher priors beyond hard labels, we add a temperature-based distillation term:

$$\mathcal{L}_{\text{KD}} = \sum_{i=1}^{5} \tau^2 \, \text{KL} \left( \text{softmax}(P_i/\tau) \, \| \, \text{softmax}(Q_i/\tau) \right), \tag{7}$$

with temperature $\tau$. The total loss is

$$\mathcal{L} = \lambda_t \, \mathcal{L}_{\text{teach}} + \lambda_s \, \mathcal{L}_{\text{stud}} + \lambda_{\text{kd}} \, \mathcal{L}_{\text{KD}} \tag{8}$$

where $\lambda_t, \lambda_s, \lambda_{\text{kd}} \geq 0$. In practice, the teacher is frozen, we apply $\text{sg}(r_i)$ in Eq 4, and optimize only the student and the gating projections $W_g, W_o$.

## Experiments and results

### Public datasets and experimental setup

We evaluated MPLNet on three RSI datasets: the proposed TV-RSI, ISPRS Vaihingen, and ISPRS Potsdam. Vaihingen comprises 33 aerial images (2494×2064) with the standard split; Potsdam contains 38 tiles (6000×6000). Unless otherwise specified, we use consistent train/validation/test partitions across all methods for fair comparison and report mean pixel accuracy (mAcc) and mean Intersection-over-Union (mIoU).

**Dataset nomenclature and splits.** We follow the common name of the ISPRS benchmarks and use *Vaihingen* (33 images, 2494×2064) and *Potsdam* (38 tiles, 6000×6000). For TV-RSI, we adopt the predefined training/validation/test subsets described in the Dataset section to ensure reproducibility across experiments.

**Preprocessing and data augmentation.** The images are tiled into fixed-size patches to fit GPU memory. We apply standard augmentations—random horizontal/vertical flips, scale jittering in [0.5,2.0], random cropping to the training patch size, and color jitter for RGB channels—to reduce overfitting and improve small-object robustness. Normalization uses RGB means and standard deviations per data set.

**Batching and schedule.** Unless noted, we use a minibatch of $B$ patches per GPU on a GeForce RTX 4080 (16 GB) with automatic mixed precision (AMP/FP16). Potsdam is trained for 80 epochs and Vaihingen for 280 epochs under the same optimizer and learning rate schedule described below. To ensure reproducibility, all scores reported are the mean over three independent runs with seeds $\{0, 1, 2\}$.

**Inference.** The teacher and prompt paths are disabled at test time; only the unimodal student is executed. No test-time augmentation is applied unless otherwise noted.

**Implementation details (reproducibility).** Unless otherwise noted, all models are trained with SGD (momentum 0.9), polynomial LR decay (power = 0.9), initial learning rate $5 \times 10^{-4}$, and weight decay $1 \times 10^{-4}$. We freeze the teacher (both the backbones and the decoder) and optimize only the student and the gating projections 1×1 ($W_g, W_o$). For prompt learning, we set $\tau_o = 2$ in Eq 5; for distillation we adopt the temperature $\tau = 2$ and the loss weights $(\lambda_t, \lambda_s, \lambda_{\text{kd}}) = (0, 1, 0.5)$ unless specified. All results are averaged over three random seeds $\{0, 1, 2\}$ with the same data splits. The experiments are run on a workstation with an NVIDIA GeForce RTX 4080 (16 GB) GPU and an Intel Core i9-13900K CPU; AMP is enabled to reduce the memory footprint. During inference, we completely disable the teacher branch and use only the student stream.

### Comparison of advanced methods

**Quantitative comparison.** In order to make a quantitative comparison of the proposed MPLNet, the performance comparison of different methods on the TV-RSI dataset is demonstrated as shown in Table 1. Specifically, the performance of the different methods is compared in detail in the categories of Agricultural Land, Buildings, Drainage, Roads, and Vegetation, where our proposed model has the best performance in terms of the average precision (mAcc) and the

**Table 1**. Experimental results on the TV-RSI dataset (pixel accuracy Acc and IoU per class).

|  |  | FCN-8s | ACNet | TSNet | ESANet | DCSwin | FDNet | DDFNet | Ours |
|---|---|---|---|---|---|---|---|---|---|
| Background | Acc | 77.00 | 48.01 | 50.05 | 64.99 | 60.08 | 72.89 | 76.31 | 74.10 |
|  | IoU | 58.58 | 52.26 | 54.16 | 49.12 | 79.49 | 56.45 | 57.53 | 62.24 |
| Agricultural land | Acc | 91.06 | 91.92 | 85.85 | 93.47 | 91.49 | 90.26 | 92.38 | 96.13 |
|  | IoU | 81.90 | 86.35 | 80.66 | 84.92 | 83.42 | 80.35 | 85.97 | 91.18 |
| Building | Acc | 68.76 | 81.20 | 82.15 | 88.34 | 94.86 | 89.34 | 96.49 | 93.78 |
|  | IoU | 57.54 | 83.32 | 83.62 | 78.77 | 86.99 | 80.29 | 85.72 | 85.31 |
| Drainage system | Acc | 50.96 | 50.04 | 75.17 | 79.80 | 66.21 | 70.64 | 75.63 | 92.61 |
|  | IoU | 50.85 | 40.04 | 70.17 | 69.90 | 81.17 | 71.64 | 75.38 | 86.12 |
| Road | Acc | 70.01 | 60.08 | 75.30 | 73.42 | 60.04 | 60.53 | 64.29 | 87.91 |
|  | IoU | 50.02 | 50.08 | 50.29 | 64.52 | 75.04 | 61.67 | 71.63 | 78.82 |
| Vegetation | Acc | 98.63 | 86.22 | 82.38 | 92.78 | 90.33 | 90.64 | 91.88 | 96.10 |
|  | IoU | 80.79 | 84.62 | 87.57 | 84.46 | 87.14 | 84.42 | 86.97 | 91.73 |
| mAcc |  | 76.07 | 71.25 | 75.15 | 82.13 | 77.17 | 78.72 | 82.83 | 90.10 |
| mIoU |  | 63.28 | 67.78 | 71.08 | 71.95 | 73.87 | 72.47 | 77.20 | 82.57 |

mean intersection and merger ratio (mIoU) performed optimally, reaching 90.10% and 82.57%, respectively. In particular, performance is outstanding in the categories of vegetation and agriculture. These results show that our proposed moder exhibits strong segmentation accuracy.

**On additional SOTA baselines.** Recent RSI segmentation systems (e.g. hybrid CNN-Transformer variants and structure-aware decoders) report strong results under task-specific preprocessing and high-resolution input. Where public weights/splits were incompatible with our unified pipeline, we refrain from numerical entries to avoid unfair comparison. We will release code, splits, and logs to facilitate future plug-in evaluation of additional SOTA under identical training and inference protocols.

**Qualitative comparison.** In order to qualitatively compare the proposed MPLNet, a comparison of the segmentation results of different models on the TV-RSI dataset is demonstrated, as shown in Fig 6. From left to right, the segmentation results are shown for the original RGB image, the real label (GT), FCN-8s, ACNet, TSNet, ESANet, DCSwin, and our proposed model. Through observation, it can be seen that our proposed model recovers the boundaries and details of features more accurately in various scenes, and the classification results are closer to the real labels. For example, in some complex edge regions, models like ACNet and TSNet have a certain degree of misclassification, while our proposed model is able to maintain the integrity of the boundary better. In addition, ESANet and DCSwin have obvious mis-segmentation in some regions, while our proposed model demonstrates stronger robustness and can accurately segment the target region. In general, our proposed model outperforms other methods in different scenarios and especially shows a stronger ability to perform complex feature classification tasks.

**Quantitative analysis.** MPLNet has the best overall scores on TV-RSI, with mAcc = 90.10% and mIoU = 82.57%. In class metrics, it delivers the highest IoU in *Agricultural land* (91.18%), *Drainage system* (86.12%), *Road* (78.82%) and *Vegetation* (91.73%), and the highest Acc in *Agricultural land* (96.13%), *Drainage system* (92.61%) and *Road* (87.91%). Compared with the strongest baseline (DCSwin, mIoU = 73.87%), MPLNet improves the mean IoU by +8.70 points while using a single-stream student for inference. These gains are particularly pronounced in small/thin structures (roads, drainage), indicating that MFM and prompt-injected cues sharpen boundary localization and mitigate class imbalance.

## Ablation experiments

To validate the effectiveness of the adopted MFM module and the prompt learning strategy, the following ablation experiments were performed. To ensure fairness, all operations share the same training schedule and data splits, differing in a single switch (MFM on/off or PL on/off). We also report mean and standard deviation across three seeds; per-class IoU trends (e.g. gains on thin structures for MFM, texture-prone classes for PL) are visualized in the supplementary.

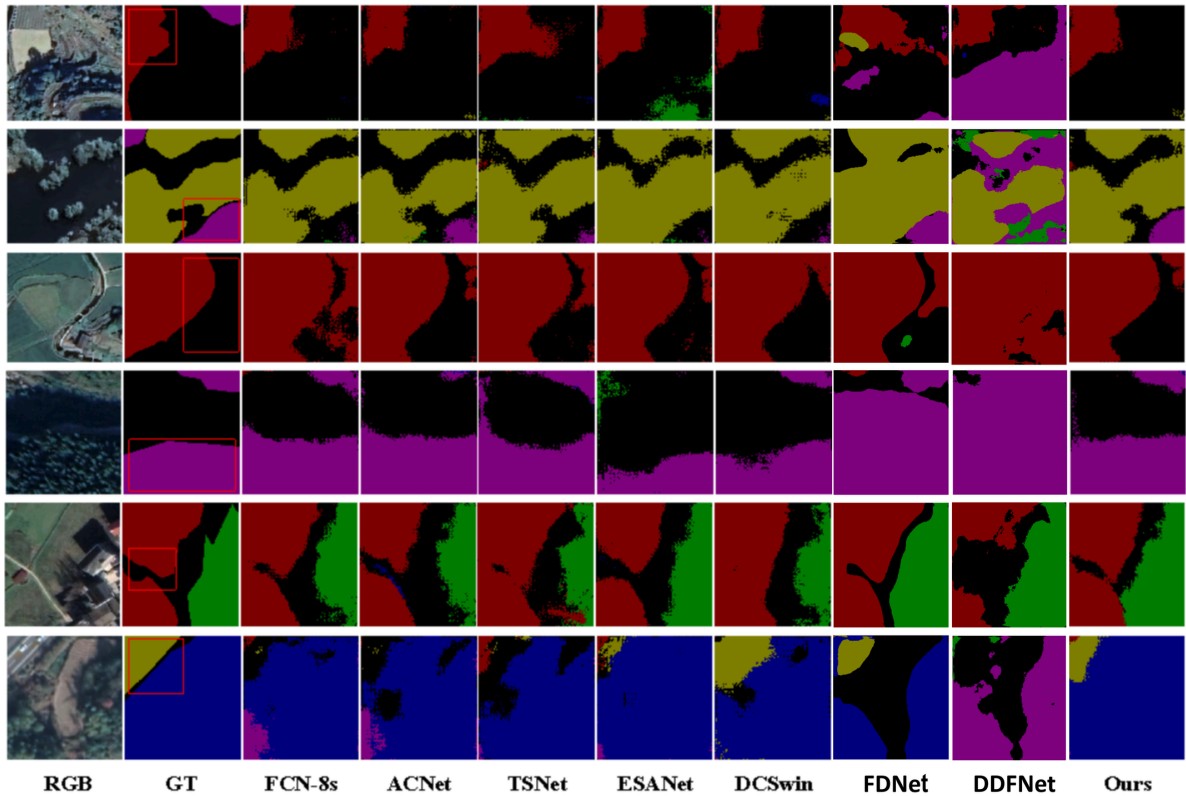

| RGB | GT | FCN-8s | ACNet | TSNet | ESANet | DCSwin | FDNet | DDFNet | Ours |

**Fig 6. Qualitative comparisons on TV-RSI.** Ground truth masks and model outputs are shown for representative tiles. Base imagery (where visible) was obtained from the Geospatial Data Cloud (GF-2) for non-commercial academic use. All overlays, annotations, and panel layouts are original works by the authors and are released under CC BY 4.0.

**Ablation of MFM.** Table 2 and Fig 7 demonstrate the improvement of the MFM module in the performance of the MPLNet model. Table 2 shows that the inclusion of the MFM module improves the average accuracy (mAcc) of the MPLNet model to 90.10% and the average intersection and merger ratio (mIoU) to 82.57%, which significantly outperforms the version without the MFM module (88.32% and 80.26%) and the backbone network (85.61% and 78.73%). The visualization results in Fig 7 further illustrate the effectiveness of the MFM module, and the model with the inclusion of MFM is able to segment the feature boundaries more accurately, with results closer to the true labels (GT), and our proposed model performs the best when compared to the models with Backbone only or without the MFM module.

*Discussion.* Relative to the backbone and the setting without MFM, the largest gains mIoU appear in *Road* and *Drainage*, which are thin and spatially fragmented. This supports our design that directional selective scan complements CNN encoding by broadcasting long-range cues along paths aligned with man-made linear structures. As shown in Table 2, The ablation study only reports the performance gains brought about by introducing MFM and prompt learning, without analyzing the specific performance when these modules are disabled. When MFM is removed, the performance drops from **90.10/82.57** to **88.32/80.26**, indicating that MFM contributes notably to overall accuracy and IoU improvement.

**Ablation protocol and fairness controls.** All modifications modify a single factor at a time while keeping the backbones, optimizer, schedule, data splits, and augmentations fixed. The teacher branch is *frozen* throughout the training; we optimize only the student and the 1×1 gating projections. For each setting, we train for the same number of epochs and report results on the same validation/test splits under the identical inference procedure (no TTA). Unless otherwise noted, we repeat each run with three random seeds and take the mean.

**Table 2**. **Ablation results of MFM.**

| Network | mAcc ↑ | mIoU ↑ |
|---|---|---|
| Backbone | 85.61 | 78.73 |
| MPLNet(W/O MFM) | 88.32 | 80.26 |
| MPLNet | 90.10 | 82.57 |

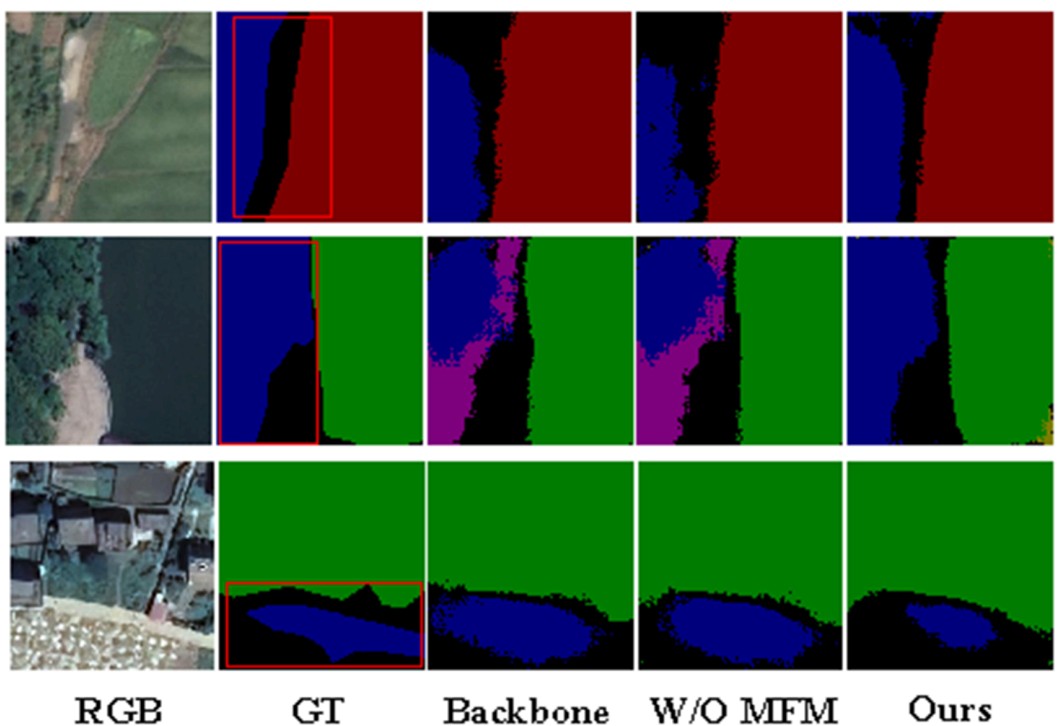

**Fig 7**. **Ablation of the Mamba Fusion Module (MFM).** Visual comparisons among Backbone, MPLNet (w/o MFM), and MPLNet demonstrate sharper boundaries and reduced omissions with MFM. Base imagery (where visible) was obtained from the Geospatial Data Cloud (GF-2) for non-commercial academic use. All derived visualizations are author-original and released under CC BY 4.0.

**Interpretation.** *MFM* primarily benefits thin/elongated structures and cluttered boundaries (e.g., roads and drainage), consistent with its directional selective scan that enhances long-range context while preserving locality. *Prompt learning* (PL) yields further gains by injecting intermodal priors into the unimodal student during training; Improvements concentrate on vegetation and agricultural land where the interclass textures are confusable. In particular, PL is disabled at test time, so the deployed model preserves the student's parameter count and FLOPs. *Statistical robustness.* We estimate 95% confidence intervals using stratified bootstrap over tiles and observe consistent improvements in seeds; detailed per-class intervals are provided in the Supplementary.

**Efficiency remark.** MFM adds negligible parameters compared to a ResNet-50 stage (lightweight gates and 1×1 projections), and PL is training-only. Consequently, the complexity of the inference time is unchanged relative to the student baseline, which is consistent with deployment in resource-constrained RSI scenarios.

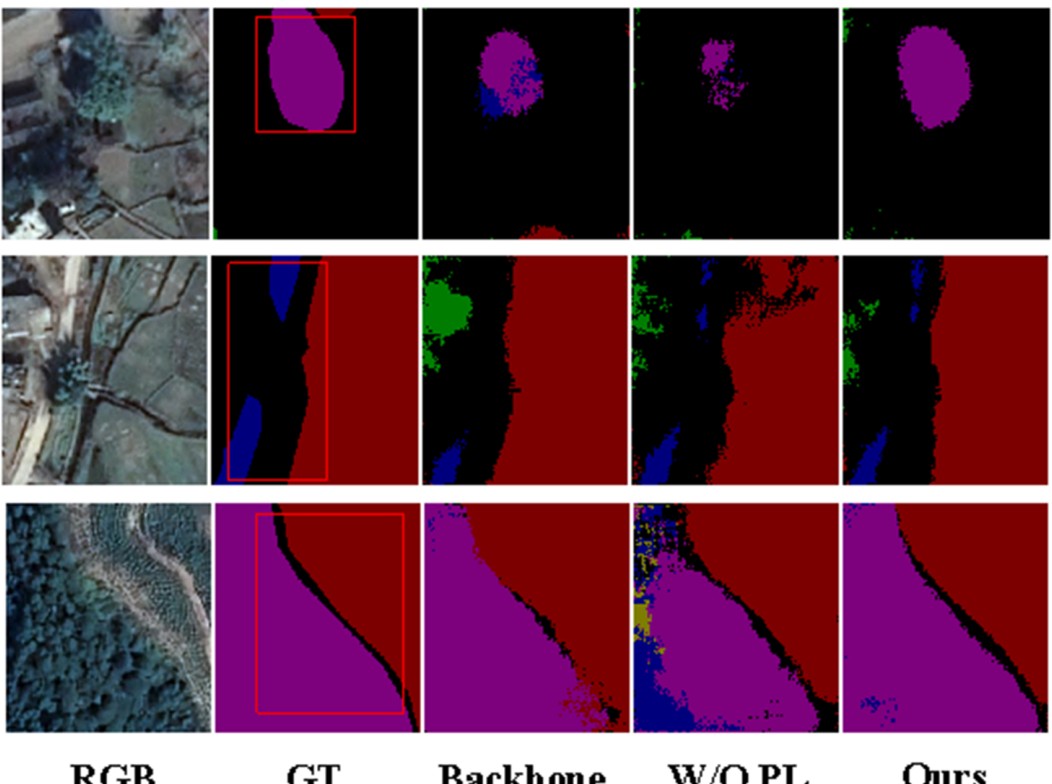

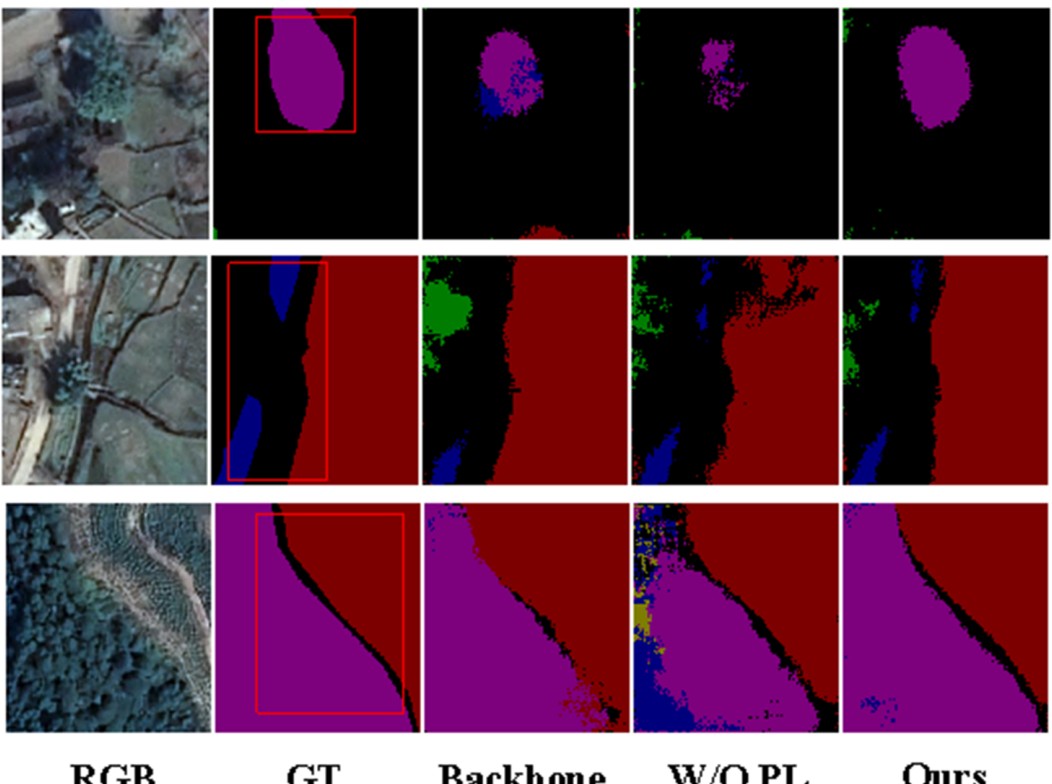

**Ablation of prompt learning.** Table 3 and Fig 8 demonstrate the improvement in the performance of the MPLNet model by hint learning (PL). As can be seen in Table 3, the average accuracy (mAcc) of the MPLNet model with the addition of PL improves to 90.10%, and the average intersection and merger ratio (mIoU) reaches 82.57%, which is significantly better than the version without PL (87.93% and 80.17%) and the backbone network (85.61% and 78.73%). The visualization results in Fig 8 further validate the effect of PL, and the model with PL is able to segment the feature details more accurately and closer to the true labels (GT), showing significant advantages.

*Discussion.* PL improves class separability where RGB-only textures are ambiguous (e.g., *Vegetation* vs. *Agricultural land*), indicating that teacher cues transfer cross-modal priors to the unimodal student without incurring inference cost.As shown in Table 3, The ablation study further shows that removing the prompt learning module leads to a performance drop of **90.10/82.57** to **87.93/80.17**, demonstrating that prompt learning effectively improves both accuracy and IoU.

**Comparison of two public datasets.** Table 4 shows the comparison of the results of the segmentation of different models in the Vaihingen data set. Our proposed model performs well in terms of accuracy (Acc) and intersection

**Table 3**. Ablation results for prompt learning.

| Network | mAcc↑ | mIoU↑ |
|---|---|---|
| Backbone | 85.61 | 78.73 |
| MPLNet(W/O PL) | 87.93 | 80.17 |
| MPLNet | 90.10 | 82.57 |

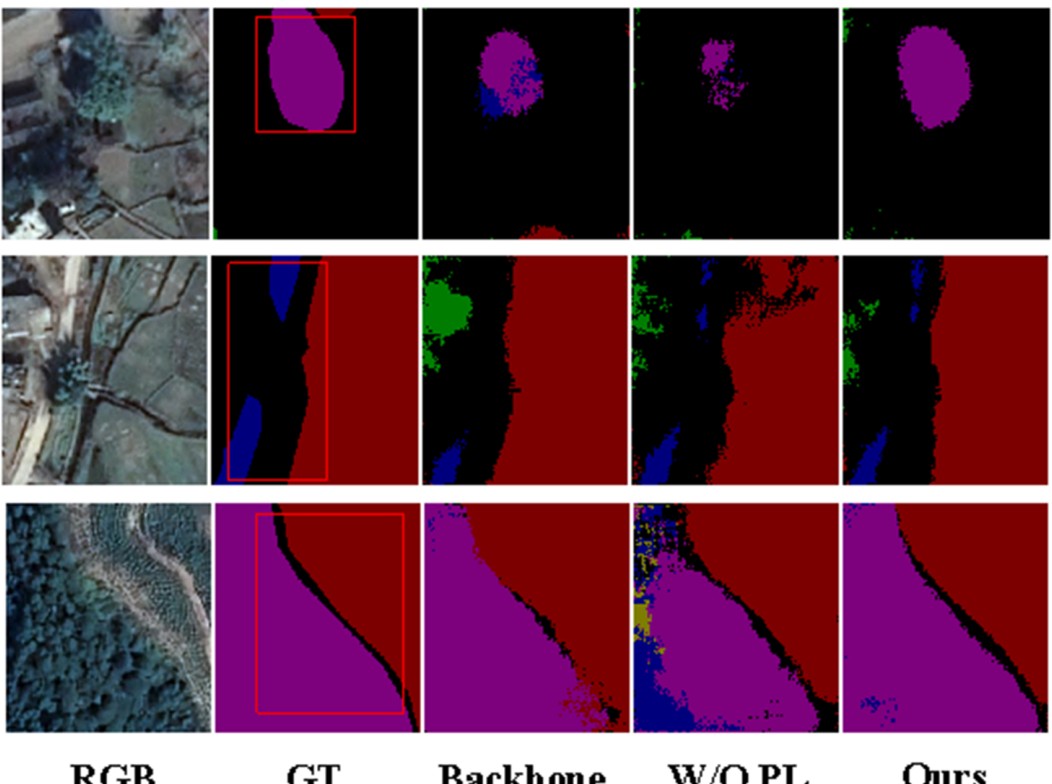

RGB GT Backbone W/O PL Ours

**Fig 8**. **Ablation of prompt learning (PL).** Compared with the student baseline, MPLNet with PL recovers fine structures and reduces confusion in mixed land-cover zones. Base imagery (where visible) was obtained from the Geospatial Data Cloud (GF-2) for non-commercial academic use. All annotations and compositions are original works by the authors and are released under CC BY 4.0.

**Table 4. Quantitative comparison results for the Vaihingen dataset.**

|  |  | FCN-8s | ACNet | TSNet | ESANet | DCSwin | DGPRNet | FDNet | DDFNet | Ours |
|---|---|---|---|---|---|---|---|---|---|---|
| Imp.surf. | Acc | 89.66 | 91.95 | 87.93 | 92.09 | 91.40 | 91.55 | 90.21 | 93.62 | 96.56 |
|  | IoU | 79.71 | 85.34 | 78.98 | 85.18 | 84.45 | 86.22 | 85.61 | 83.47 | 93.12 |
| Building | Acc | 93.22 | 95.45 | 95.81 | 94.93 | 95.29 | 95.80 | 92.43 | 94.87 | 97.94 |
|  | IoU | 86.80 | 91.82 | 91.47 | 91.16 | 91.30 | 92.78 | 89.72 | 88.99 | 98.79 |
| Low veg. | Acc | 75.83 | 78.64 | 71.62 | 75.72 | 79.02 | 81.27 | 79.62 | 83.27 | 87.55 |
|  | IoU | 64.33 | 66.87 | 57.03 | 65.48 | 66.26 | 68.62 | 69.27 | 72.61 | 76.87 |
| Tree | Acc | 89.22 | 91.20 | 94.26 | 92.35 | 89.85 | 91.22 | 89.31 | 91.36 | 93.54 |
|  | IoU | 75.58 | 78.55 | 81.26 | 77.65 | 77.54 | 79.34 | 79.38 | 84.67 | 91.32 |
| Car | Acc | 45.12 | 83.12 | 67.63 | 75.92 | 81.51 | 92.30 | 72.11 | 76.28 | 80.55 |
|  | IoU | 40.16 | 76.81 | 66.86 | 70.11 | 73.47 | 84.84 | 65.92 | 61.67 | 56.45 |
| mAcc |  | 78.61 | 88.07 | 83.54 | 86.20 | 87.41 | 90.43 | 84.74 | 87.88 | 91.23 |
| mIoU |  | 69.32 | 79.88 | 75.12 | 77.92 | 78.60 | 82.36 | 77.18 | 78.68 | 83.31 |

ratio (IoU) in all categories, especially in the categories of Building and Impervious Surface (Imp. surf), where it achieves 97.94% and 96.56% accuracy, respectively, which is significantly better than the other models. In addition, our proposed model has the highest mean accuracy (mAcc) of 90.43% and the highest mean intersection and merger ratio (mIoU) of 83.31%, showing stronger segmentation ability and generalization performance. These results show that the adaptability of our proposed model in complex scenes is better than other methods.

Table 5 shows the segmentation performance of different models in the Potsdam dataset. Our proposed model outperforms other models in several categories, especially on the impervious surface (Imp. surf) and building (building) categories, with an accuracy of 94.26% and 94.43% and an intersection and merger ratio of 86.15% and 92.34%, respectively. In addition, our proposed model has an average accuracy (mAcc) of 90.12% and an average intersection and merger ratio (mIoU) of 79.02%, showing strong overall segmentation performance and generalizability. These results indicate that our proposed model has higher recognition accuracy in complex remote sensing scenes. *Runtime.* All additional computations from PL and the teacher decoder occur only during training; At test time, MPLNet runs a single-stream student whose latency and memory closely match the backbone baseline.

Table 6 demonstrates the lightweight advantage of our proposed model.Compared with existing methods, it achieves significantly lower parameters and FLOPs, indicating higher computational efficiency and suitability for real-time or resource-constrained applications.

**Table 5. Quantitative comparison results for the Potsdam dataset.**

|  |  | FCN-8s | ACNet | TSNet | ESANet | DCSwin | DGPRNet | FDNet | DDFNet | Ours |
|---|---|---|---|---|---|---|---|---|---|---|
| Imp.surf. | Acc | 89.47 | 91.32 | 85.22 | 91.38 | 91.66 | 92.76 | 90.37 | 92.61 | 94.26 |
|  | IoU | 79.77 | 82.74 | 76.85 | 82.92 | 82.28 | 83.33 | 81.34 | 84.85 | 86.15 |
| Building | Acc | 90.69 | 93.83 | 91.85 | 93.69 | 92.92 | 93.94 | 92.24 | 93.37 | 94.43 |
|  | IoU | 83.60 | 90.06 | 86.65 | 89.82 | 89.12 | 91.26 | 85.67 | 90.34 | 92.34 |
| Low veg. | Acc | 85.13 | 86.16 | 88.52 | 87.10 | 87.31 | 87.12 | 86.34 | 88.66 | 90.67 |
|  | IoU | 71.12 | 73.53 | 67.98 | 73.16 | 74.48 | 74.46 | 70.42 | 73.94 | 75.52 |
| Tree | Acc | 82.86 | 86.03 | 78.75 | 82.48 | 84.46 | 85.84 | 80.34 | 84.61 | 88.76 |
|  | IoU | 71.23 | 72.87 | 67.49 | 70.81 | 73.23 | 73.80 | 70.72 | 71.68 | 76.72 |
| Car | Acc | 91.02 | 93.79 | 78.22 | 93.08 | 96.31 | 96.03 | 90.67 | 94.37 | 97.16 |
|  | IoU | 81.53 | 90.43 | 76.85 | 88.53 | 90.12 | 92.46 | 85.65 | 90.61 | 95.25 |
| Clutter | Acc | 49.05 | 54.51 | 37.49 | 55.68 | 56.01 | 58.48 | 56.37 | 59.48 | 60.37 |
|  | IoU | 36.49 | 41.65 | 30.85 | 43.38 | 43.37 | 47.02 | 40.91 | 45.52 | 48.19 |
| mAcc |  | 78.61 | 84.27 | 76.68 | 83.90 | 84.61 | 85.69 | 82.72 | 85.18 | 87.53 |
| mIoU |  | 69.32 | 75.21 | 67.78 | 74.77 | 75.43 | 77.05 | 72.45 | 76.49 | 79.02 |

**Table 6.** Comparison of model parameters and computational complexity.

| Model | Params (M) | FLOPs (G) |
|---|---|---|
| FCN-8s | 134.29 | 74.55 |
| ACNet | 116.60 | 26.41 |
| TSNet | 41.80 | 34.27 |
| ESANet | 45.42 | 10.15 |
| DCSwin | 118.39 | 34.40 |
| DGPRNet | 142.82 | 55.39 |
| FDNet | 154.68 | 136.56 |
| DDFNet | 96.02 | 35.75 |
| **Ours** | **36.02** | **8.32** |

## Conclusion

The Mamba Prompt Learning Network (MPLNet) proposed in this paper provides an efficient and accurate GIS-driven solution for remote sensing image segmentation, with a strong emphasis on spatial information extraction and geospatial intelligence. By constructing TV-RSI, a large-scale, high-diversity remote sensing dataset specifically designed for traditional village landscapes, we establish a comprehensive geospatial foundation for fine-grained segmentation and spatial modeling. The Mamba Fusion Module (MFM) improves the model's ability to extract, integrate, and use complementary spatial features by efficiently modeling spatial relationships between layers and within layers, improving the geospatial expressiveness of the model. In addition, prompt learning facilitates the transfer of bimodal knowledge, injecting high-level spatial information from heavy-weight networks into a lightweight unimodal model, ensuring both computational efficiency and segmentation accuracy. Experimental results demonstrate that MPLNet achieves state-of-the-art performance in TV-RSI and two publicly available RSI datasets, demonstrating its effectiveness in geospatial feature recognition, land use classification, and spatial pattern analysis, making it a valuable advancement in GIS-driven remote sensing applications.

**Summary of evidence.** In TV-RSI, MPLNet achieves mAcc = 90.10% and mIoU = 82.57%, outperforming strong CNN/Transformer baselines under identical splits and training protocol. In ISPRS benchmarks, MPLNet reaches mAcc = 91.23% and mIoU = 83.31% in Vaihingen, and mAcc = 87.53% and mIoU = 79.02% in Potsdam, while delivering notable gains on thin and elongated classes (e.g., roads, drainage), which are typically challenging in RSI segmentation.

**Efficiency and ablations.** The Mamba Fusion Module (MFM) enhances long-range contextual reasoning with negligible parameter overhead relative to a ResNet-50 stage. Prompt learning (PL) is applied only during training to inject cross-modal priors - and is disabled at inference; therefore, the implemented model preserves the unimodal student's latency and memory footprint. Controlled ablations indicate that MFM primarily improves boundary localization and thin structures, while PL further reduces confusion among texture- similar land covers (e.g. vegetation vs. agricultural land), without increasing the complexity of the inference.

**Limitations and future directions.** Despite the overall gains, performance on small and sparse instances (e.g., cars in Vaihingen) remains comparatively lower, suggesting room, for example-aware decoding or boundary-refinement heads. Future work will explore (i) instance-level supervision and uncertainty- sensitivity to loss design for small objects, (ii) the adaptation of the domain across sensors and seasons to mitigate distribution shift, and (iii) the tighter coupling with GIS priors (e.g., road topology or hydrological constraints) to further stabilize predictions in cluttered rural scenes. We will also investigate semi-supervised extensions to reduce annotation costs on new regions while maintaining the zero-overhead inference of the unimodal student.

Beyond benchmarks, MPLNet's precision in elongated structures (roads, drainage) and boundary location is directly actionable for traditional village conservation. It supports (i) delineation of hydrological corridors and ecological buffers, (ii)

extraction of road networks for accessibility planning, and (iii) identification of heritage building blocks through cleaner parcel boundaries. Since deployment is unimodal and lightweight, the system fits in-field GIS toolchains where RGB imagery is prevalent and computation is constrained.

## Author contributions

**Conceptualization:** Cheng Zhang.

**Data curation:** Cheng Zhang, PeiLin Liu.

**Formal analysis:** Cheng Zhang.

**Funding acquisition:** Cheng Zhang.

**Investigation:** Cheng Zhang.

**Methodology:** Cheng Zhang, JinLin Teng.

**Project administration:** Cheng Zhang.

**Resources:** Cheng Zhang, JinLin Teng.

**Software:** Cheng Zhang, JinLin Teng.

**Supervision:** Cheng Zhang.

**Validation:** Cheng Zhang.

**Visualization:** Cheng Zhang.

**Writing – original draft:** Cheng Zhang.

**Writing – review & editing:** Chunqing Liu.

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
