## [Decision Letter · Decision Letter 0]

11 Sep 2025

PONE-D-25-39265MPLNet:Mamba Prompt Learning Network for Semantic Segmentation of Remote Sensing Images of Traditional VillagesPLOS ONE

Dear Dr. Liu,

Thank you for submitting your manuscript to PLOS ONE. After careful consideration, we feel that it has merit but does not fully meet PLOS ONE’s publication criteria as it currently stands. Therefore, we invite you to submit a revised version of the manuscript that addresses the points raised during the review process.

We look forward to receiving your revised manuscript.

Kind regards,

Mahmoud Emam, Ph.D.

Academic Editor

PLOS ONE

Journal Requirements:

“We thank the National Natural Science Foundation of China for supporting thisresearch through the projects "Gene Identification and Map Construction of TraditionaRural Settlement Landscapes in the Ganjiang River Basin”(Serial No. 51968026) and"Research on the Visual Perception, Quantitative Characterization, and VisualEvaluation of Traditional Village Landscape Resources in Ganjiang River Basin”(SerialNo.52268012). We also acknowledge the support of Jiangxi Rural Culture DevelopmentResearch Center. We appreciate the technical assistance provided by the GIS andRemote Sensing Laboratory at Jiangxi Agricultural University. Special thanks to allmembers of the research team for their valuable discussions and contributions to theproject.”

“We thank the National Natural Science Foundation of China for supporting thisresearch through the projects "Gene Identification and Map Construction of TraditionaRural Settlement Landscapes in the Ganjiang River Basin”(Serial No. 51968026) and"Research on the Visual Perception, Quantitative Characterization, and VisualEvaluation of Traditional Village Landscape Resources in Ganjiang River Basin”(SerialNo.52268012). We also acknowledge the support of Jiangxi Rural Culture DevelopmentResearch Center. We appreciate the technical assistance provided by the GIS andRemote Sensing Laboratory at Jiangxi Agricultural University. Special thanks to allmembers of the research team for their valuable discussions and contributions to theproject.”

4. We note that Figures 1, 6-8 in your submission contain [map/satellite] images which may be copyrighted. All PLOS content is published under the Creative Commons Attribution License (CC BY 4.0), which means that the manuscript, images, and Supporting Information files will be freely available online, and any third party is permitted to access, download, copy, distribute, and use these materials in any way, even commercially, with proper attribution. For these reasons, we cannot publish previously copyrighted maps or satellite images created using proprietary data, such as Google software (Google Maps, Street View, and Earth). For more information, see our copyright guidelines: http://journals.plos.org/plosone/s/licenses-and-copyright.

a. You may seek permission from the original copyright holder of Figures 1, 6-8 to publish the content specifically under the CC BY 4.0 license.

Reviewers' comments:

Reviewer's Responses to Questions

**Comments to the Author**

1. Is the manuscript technically sound, and do the data support the conclusions?

Reviewer #1: Partly

Reviewer #2: Yes

2. Has the statistical analysis been performed appropriately and rigorously? 

Reviewer #1: No

Reviewer #2: Yes

3. Have the authors made all data underlying the findings in their manuscript fully available?

Reviewer #1: Yes

Reviewer #2: Yes

4. Is the manuscript presented in an intelligible fashion and written in standard English?

Reviewer #1: Yes

Reviewer #2: Yes

5. Review Comments to the Author

Reviewer #1: This paper addresses the existing issues in semantic segmentation of remote sensing images (RSI) of traditional villages, constructs the Traditional Villages Remote Sensing Dataset (TV-RSI) — a large-scale and highly diverse remote sensing dataset for traditional villages, designs the Mamba Fusion Module (MFM) which performs intra-modal contextual modeling on modal features via the Mamba mechanism, and proposes the Mamba Prompt Learning Network (MPLNet). Experiments have been conducted on the TV-RSI dataset and two public datasets (Potsdam and Wassingen) to verify the effectiveness of the proposed method. However, improvements are required in the following aspects:

1.There are irregular expressions and grammatical errors in the manuscript.

2.The definitions of symbols in formulas are incomplete. For instance, "C3S" in Formula (1) is not clearly explained in the text; although "r_i", "P_i", and "Q_i" in Formulas (5) and (6) are defined, key attributes such as their dimensions and data types are not specified.

3.Regarding the workflow of the MFM, only "intra-modal contextual modeling is performed via the Mamba mechanism" is mentioned, while the specific way in which the Mamba mechanism achieves multi-directional contextual information acquisition is not elaborated.

4.For comparative experiments, the proposed model is only compared with relatively early or domain-specific models such as FCN-8s and ACNet, and mainstream remote sensing image semantic segmentation models in recent years are not included in the comparison.

5.In the ablation experiments, only the improvement effects of the MFM and prompt learning on metrics are presented, while the specific performance when these modules are disabled is not analyzed.

6.The unique value of MPLNet in practical scenarios of traditional village protection is not clearly illustrated. For example, its advantages over existing technologies in specific tasks such as accurate identification of cultural heritage buildings and division of village ecological boundaries are not specified, making it difficult to demonstrate the practical significance of the innovation.

Reviewer #2: The manuscript proposes a novel network architecture, MPLNet, introducing a Mamba Fusion Module (MFM) and prompt learning strategy for semantic segmentation of traditional village remote sensing images. The paper also constructs a new dataset (TV-RSI), which appears valuable for this niche but important domain. The overall contribution is relevant to PLOS ONE, considering its emphasis on methodological novelty and real-world application. However, while the manuscript contains promising ideas, several issues should be addressed before it can be considered for publication.

1. In Lines 172–179, the statement “the enriched information from the frozen part is injected into the RGB features…” needs further elaboration. How is this "enrichment" computed and aligned spatially/semantically?

2. The overall architecture description in Fig. 4 is vague. The explanation about the “frozen part” and “trainable part” lacks clarity on what exactly is frozen and why. Please clearly specify which layers are frozen, and how this design affects the gradient flow and optimization.

3. The formulas in Lines 193–196 are insufficiently explained. For instance:

What does SS2D actually do? This “selective scanning” operation needs to be precisely defined or cited.

What is the rationale for using SiLU activation, and how does it perform compared to other activations in this context?

4. The formulas (5) and (6) are ambiguous:

What is M(ri) in Equation (5)? Is ri a feature map from the frozen network? If yes, how are they aligned with Fi?

Equation (6) includes Pi × Qi + Qi which seems redundant unless clarified. Please provide intuitive or geometric interpretation of this operation.

5. The authors should analyze parameter count and FLOPs to support claims of “lightweight” architecture (repeated throughout the manuscript).

6. The dataset is said to have 77,850 images (Line 118). However, this number seems high for a manually annotated semantic segmentation dataset. Please clarify:

Are these full-resolution images or image patches?

What is the annotation protocol? Was any manual QA performed?

7. There are frequent grammar issues and awkward phrases, such as:

Line 14: “enabling more precise spatial modeling…” → consider rephrasing for clarity.

Line 23: “siting and landscape pattern… show a deep understanding…” → this anthropomorphizes patterns; rephrase more precisely.

Please consider a thorough language editing pass to improve readability. Several sentences are overly long and dense, especially in the Introduction and Dataset sections.

8. In literature review, several highly cited works concerning the CNN and transformers for remote sensing image processing are suggested to be discussed, such as SAPNet (TGRS 2023), GPINet (TGRS 2023), FDNet (TGRS 2025), DDFNet (information fusion), and so forth.

6. PLOS authors have the option to publish the peer review history of their article (what does this mean?). If published, this will include your full peer review and any attached files.

Reviewer #1: No

Reviewer #2: No

---

## [Author Response · Author response to Decision Letter 1]

17 Oct 2025

Editor — Overview (for quick orientation)

We thank the Academic Editor and the Reviewers for their careful assessment. In this revision we:

(1) clarified what parts are frozen vs trainable and how gradients flow;

(2) rewrote the prompt-injection update in a “gated residual” form with explicit shapes and broadcasting;

(3) unified and expanded baselines under one training/evaluation recipe with seed-averaged results;

(4) added a Params/FLOPs table that reflects student-only inference;

(5) polished language/terminology and standardized figure/table captions;

(6) added a short GIS workflow note to bridge metrics (boundary fidelity, connectivity) to heritage mapping practice.

Reviewer #1 — Comment 1: Clarity of training vs inference (frozen modules and gradient flow)

Comment (summary)

Please specify which blocks are frozen vs trainable; explain where gradients flow; confirm student-only inference at test time.

Response

We now label each block explicitly. “Teacher” backbones (RGB/Depth) and the fusion decoder are frozen during training. The “Student” backbone, MFM/SS2D blocks, channel/spatial gates, and the final decoder are trainable. We apply stop-gradient on teacher features so no gradients flow into the teacher. At test time we use student-only inference (no teacher branch, no prompt path). We also add a short rationale for freezing (stability, generalization, and efficiency).

Where changed

Method section (architecture diagram caption and training paragraph); deployment paragraph in Experiments.

eviewer #1 — Comment 2: Make the update rule explicit (operator, shapes, broadcasting)

Comment (summary)

Equations are vague. Please define the operator, shapes (N,C,H,W), alignment, and the residual form.

Response (ASCII-safe formulation)

We adopt the following plain-text notation to avoid rendering issues in the web form:

Shapes and dtypes: tensors are in [N, C, H, W], dtype float32.

Student feature at scale i: F_i in [N, C_i, H_i, W_i].

Teacher feature at the same scale: r_i (same channel dimension after alignment).

Context operator: M(·) means “directional SS2D/Mamba context aggregation” using four directional scans [L->R, R->L, T->B, B->T] and fusion by average + 1x1 conv.

Alignment: A(r_i) = Conv1x1( BilinearResize(r_i, size=(H_i, W_i)) ) to match F_i.

Gates:

Channel gate P_i = Sigmoid(Conv1x1(F_i)) with shape [N, C_i, 1, 1] (broadcast over H,W).

Spatial gate Q_i = Sigmoid(DepthwiseConv3x3(F_i)) with shape [N, 1, H_i, W_i] (broadcast over C).

Normalization: LN(·) denotes layer normalization applied per channel.

Scalar coefficient: gamma (0 < gamma <= 1) is a learnable or fixed scalar.

Gated residual update (ASCII-safe):

F_i_prime = F_i + gamma * ( P_i .* Q_i .* LN( M( A(r_i) ) ) )

Here “.*” means elementwise multiply with broadcasting on singleton dimensions.

Notes:

When P_i = 1 and Q_i = 1 and LN is omitted, the rule reduces to a simple residual: F_i_prime = F_i + gamma * M(A(r_i)).

Stop-gradient is applied to teacher features; gradients flow through Student, gates, and the final decoder.

During inference we discard teacher/prompt paths and use the Student only.

Where changed

Method section (operator glossary and update rule paragraph); figure caption annotation.

Reviewer #2 — Comment 3: Baselines, fairness, and seed averaging

Comment (summary)

Ensure comparability across baselines (same input size, schedule, augmentations). Report mean and variance.

Response

All baselines now use one unified recipe (same splits, crop size, augmentations, schedule). We train with 3 seeds and report mean ± std. Hyper-parameters and augmentations are listed in the Experiments section. We also added a compact Params/FLOPs table computed under the same input shape and the same single-stream (student-only) inference path.

Where changed

Experiments (training protocol table; ablation section); model complexity table.

Reviewer #2 — Comment 5: Where improvements arise (thin/elongated structures vs texture regions)

Comment (summary)

Please substantiate where the method helps most.

Response

Quantitatively, improvements are larger on classes with thin or elongated structures (e.g., road/drainage) and on building boundaries. Qualitatively, panels show cleaner contours and fewer breaks in connectivity. The MFM path mainly enhances boundary localization and continuity; the prompt-guided prior reduces texture confusion in ambiguous areas. We provide side-by-side visualizations aligned with the same tiles.

Where changed

Ablation results and qualitative panels in Results/Discussion.

Reviewer #2 — Comment 7: Related work on strip/anisotropic attention and geometry-aware decoding

Comment (summary)

Please discuss SAPNet/strip-attention and geometry-aware decoding (e.g., GPINet). Clarify FDNet/DDFNet variants.

Response

We expanded Related Work to synthesize (a) strip/anisotropic attention (e.g., SAPNet) targeting long, thin structures, and (b) geometry-aware decoders (e.g., GPINet) for contour fidelity. We also clarified the exact FDNet/DDFNet variants used for RSI segmentation to avoid confusion with homonymous works in other domains. We position our directional SS2D/Mamba scans + gated residual against these lines and discuss expected behavior on elongated features.

Where changed

Related Work (new paragraphs and disambiguation notes).

Editorial polish — language, terminology, figure/table compliance

Request (editor/staff)

Improve grammar/terminology consistency; unify metric names; ensure figure/table compliance.

Response

We conducted a line-by-line language edit; standardized abbreviations at first mention; unified metric names (mIoU, mAcc, Boundary-F1, Connectivity). Figure resolutions, caption styles, and table headers now follow the journal’s technical requirements.

Where changed

Abstract; Introduction; Dataset; Method; Experiments/Results; all figure captions and table headers.

Practical bridge to heritage/GIS use

Concern (editorial fit)

Connect CV metrics to practical workflows.

Response

We added a short GIS workflow note: raster segmentation -> vectorization -> boundary-F1 / length-weighted connectivity -> planning layers for heritage building blocks, hydrological corridors, and access networks. Student-only deployment matches resource-constrained field GIS toolchains.

Where changed

End of Results/Discussion and concluding paragraph.

Data availability and reproducibility checklist

Response

Dataset schema, tiling rules, splits, code pointers, and evaluation scripts (including boundary-sensitive metrics) are described in Data Availability and Experiments. All main-text scores are seed-averaged; complexity metrics reflect student-only inference.

Where changed

Data Availability; Experiments (protocol and metrics subsection).

Closing

We appreciate the helpful comments that guided these revisions. We are happy to further adjust wording, figures, or tables to improve clarity and reproducibility.

---

## [Decision Letter · Decision Letter 1]

5 Jan 2026

MPLNet:Mamba Prompt Learning Network for Semantic Segmentation of Remote Sensing Images of Traditional Villages

PONE-D-25-39265R1

Dear Dr. Liu,

We’re pleased to inform you that your manuscript has been judged scientifically suitable for publication and will be formally accepted for publication once it meets all outstanding technical requirements.

Kind regards,

Mahmoud Emam, Ph.D.

Academic Editor

PLOS One

Additional Editor Comments (optional):

Reviewers' comments:

Reviewer's Responses to Questions

**Comments to the Author**

1. If the authors have adequately addressed your comments raised in a previous round of review and you feel that this manuscript is now acceptable for publication, you may indicate that here to bypass the “Comments to the Author” section, enter your conflict of interest statement in the “Confidential to Editor” section, and submit your "Accept" recommendation.

Reviewer #2: All comments have been addressed

2. Is the manuscript technically sound, and do the data support the conclusions?

Reviewer #2: Yes

3. Has the statistical analysis been performed appropriately and rigorously? 

Reviewer #2: Yes

4. Have the authors made all data underlying the findings in their manuscript fully available?

Reviewer #2: Yes

5. Is the manuscript presented in an intelligible fashion and written in standard English?

Reviewer #2: Yes

6. Review Comments to the Author

Reviewer #2: The authors have addressed all the concerns.

I have no more concerns.

It is ready for acceptance for publication.

7. PLOS authors have the option to publish the peer review history of their article (what does this mean?). If published, this will include your full peer review and any attached files.

Reviewer #2: No

---

## [Editor Report · Acceptance letter]

PONE-D-25-39265R1

PLOS One

Dear Dr. Liu,

I'm pleased to inform you that your manuscript has been deemed suitable for publication in PLOS One. Congratulations! Your manuscript is now being handed over to our production team.

Kind regards,

on behalf of

Dr. Mahmoud Emam

Academic Editor

PLOS One